# The Role of the RANKL/RANK Axis in the Prevention and Treatment of Breast Cancer with Immune Checkpoint Inhibitors and Anti-RANKL

**DOI:** 10.3390/ijms21207570

**Published:** 2020-10-14

**Authors:** Aristofania Simatou, Panagiotis Sarantis, Evangelos Koustas, Athanasios G. Papavassiliou, Michalis V. Karamouzis

**Affiliations:** Molecular Oncology Unit, Department of Biological Chemistry, Medical School, National and Kapodistrian University of Athens, 11527 Athens, Greece; ariasimatou@yahoo.com (A.S.); panayotissarantis@gmail.com (P.S.); vang.koustas@gmail.com (E.K.)

**Keywords:** RANK, RANKL, EGFR, ERBB2, immune checkpoint inhibitors, denosumab

## Abstract

The receptor activator of nuclear factor-κB (RANK) and the RANK ligand (RANKL) were reported in the regulation of osteoclast differentiation/activation and bone homeostasis. Additionally, the RANKL/RANK axis is a significant mediator of progesterone-driven mammary epithelial cell proliferation, potentially contributing to breast cancer initiation and progression. Moreover, several studies supported the synergistic effect of RANK and epidermal growth factor receptor (EGFR) and described RANK’s involvement in epidermal growth factor receptor 2 (ERBB2)-positive carcinogenesis. Consequently, anti-RANKL treatment has been proposed as a new approach to preventing and treating breast cancer and metastases. Recently, RANKL/RANK signaling pathway inhibition has been shown to modulate the immune environment and enhance the efficacy of anti-CTLA-4 and anti-PD-1 monoclonal antibodies against solid tumors. Clinical and experimental trials have emerged evaluating RANKL inhibition as an enhancer of the immune response, rendering resistant tumors responsive to immune therapies. Trials evaluating the combinatorial effect of immune checkpoint inhibitors and anti-RANKL treatment in double-positive (RANK+/ERBB2+) patients are encouraging.

## 1. RANKL/RANK Signaling Pathway

The well-known nuclear factor-κB (RANK)/RANK ligand (RANKL)/osteoprotegerin (OPG) axis was identified at the end of the 1990s as a multipotent regulator mechanism of bone remodeling [1]. It is composed of three main components: (a) the receptor activator of nuclear factor-kB ligand (RANKL), (b) the receptor activator of nuclear factor-kB (RANK) and (c) the soluble receptor of RANKL, osteoprotegerin (OPG). RANKL, a tumor necrosis factor-alpha superfamily cytokine, was initially identified on the surface of T-cells and dendritic cells (DCs) [2]. It is a type II membrane protein encoded by the *TNFS11* gene and encountered in three isoforms due to alternative splicing of the gene [3]. RANKL1 represents the full-length molecule, while in RANKL2, a branch of the intracellular domain is missing. In RANKL3, the N-terminal fraction is deleted. It has been highlighted either as a soluble or membrane form and is the ligand of the membrane receptor RANK. Soluble RANKL (sRANKL) is derived from the membrane-bound form through alternative splicing or the proteolytic cleavage and can potentially circulate in blood [4]. RANK, a member of the tumor necrosis factor receptor (TNFR) superfamily, encoded by the gene *TNFRSF11A*), is a type I transmembrane protein, including four cysteine-rich repeat motifs and two N-glycosylation sites. The binding of these two molecules leads to the recruitment of adaptor molecules such as TNF receptor-associated factors (TRAFs), the adaptor protein TRAF6 and the activation of a plethora of signaling pathways (JNK, AKT/PKB, NF-kb, MAPK/ERK and Src) [5]. Several studies suggest that oxidative stress is a key pathogenic mechanism of osteoporosis. NRF2 partakes in bone metabolism and has a critical role, providing a balance between the plasma antioxidant and oxidant biomarkers. The expression of RANKL decreases the ratio of NRF2/KEAP1, which decreases the expression of NRF2-related enzymes and favors the increase in ROS activity, a downstream molecular signal of RANKL. NRF2 could also affect osteoclastogenesis through the expression of IL-6 [6]. In contrast, molecules with antioxidative activity, such as petunidin, which is a B-ring 5′-O-methylated derivative of delphinidin, act as bone anabolic agents [7]. Additionally, the RANKL/RANK axis is regulated by the soluble receptor osteoprotegerin (OPG) (*TNFRSF11B*), which is a soluble glycoprotein encountered as a 60 kD monomer or a 120 kD dimer. The dimerization of OPG increases its affinity to RANKL, and by binding to it, exerts an inhibitory effect on the axis [8].

In 2009, a human monoclonal antibody against RANKL, denosumab, was developed to inhibit the interaction between RANK and its ligand RANKL [9,10]. In 2011, the drug was approved for osteoporosis treatment and bone metastases in breast and prostate carcinomas [11]. Since then, denosumab has been widely used in breast cancer (BC) patients with metastatic disease and shown to be equal or superior to zoledronic acid in holding or preventing skeletal-related events (SREs) [12,13,14,15]. Recent studies emphasize the emergence of alternative therapeutic agents, such as high-dose diosgenin, which seems to affect osteoporosis by modulating the RANKL/OPG ratio [16].

## 2. The Role of the RANKL/RANK Signaling Pathway in Normal Mammary Gland Development

Beyond bone homeostasis, recent studies have pointed out the essential role of the RANKL/RANK axis in mammary gland physiology and its role as a mediator in hormone-driven epithelial proliferation through pregnancy. RANK- and RANKL-deficient mice fail to form lobuloalveolar structures during pregnancy [17]. At the same time, RANK overexpression in transgenic mice with mouse mammary tumor virus promoter (MMTV) controlled RANK, induced proliferation at midgestation and disrupted alveolar differentiation in the mammary epithelia [18].

These observations suggest that RANK’s lack of overexpression leads to impaired lobuloalveolar development and consequent lactation defects. Considering the critical role of hormones during pregnancy, several studies in mice have shown that progesterone enhanced RANKL expression in cells that are already characterized by high estrogen and progesterone receptors (ER/PR) on the cell surface [19,20,21]. When RANKL expression is specifically induced in these cells, an ordered alveolar development occurs, and the RANKL signaling pathway seems to be responsible for the primary proliferative response of the mouse mammary epithelium to progesterone for the period of mammary lactation morphogenesis [22]. The RANK-Id2-p21 axis represents two main signaling pathways activated by RANK in mammary epithelial cells. IKK-α catalyzes the phosphorylation, ubiquitination and proteasome degradation of IkBα, leading to its dissociation from NF-kB, which then migrates to the nucleus and induces the transcription of cyclin D1. On the other hand, the inhibitor of DNA-binding protein 2 (Id2) translocates into the nucleus and inhibits the expression of p21, a well-known cell cycle inhibitor. Altogether, these molecular mechanisms lead to increased proliferation and the survival of cells [23].

Progesterone treatment does not seem to affect RANKL expression in human breast cancer cells that express progesterone receptors (PR+), indicating that the progesterone-mediated regulation of the mammary epithelium is limited to normal-expressing, and not extended to RANK-expressing, cancer cells. Additionally, RANKL has been shown to exert a paracrine function, mediating ER/PR- cell proliferation and promoting the progesterone-mediated amplification of mouse mammary stem cells [24,25,26].

## 3. The Role of the RANKL/RANK Signaling Pathway in Breast Cancer

Following the observation of progesterone being a crucial regulator of RANKL expression, which leads to enhancing the proliferation of mammary epithelial and stem cells, several studies highlight the role of the RANKL/RANK axis in the progression of breast tumorigenesis and bone and lung metastases in mice through the same pathway [27,28,29].

Consequently, anti-RANKL treatment has been proposed as a new approach to preventing and treating breast cancer and metastases [30,31,32]. Analytically, the RANK signaling pathway seems to be involved in all stages of breast cancer development, from the expansion of the partition and enhancing the proliferation of epithelial cells to increasing the resistance of tumor cells to DNA thus damaging agents and promoting metastatic potential [17,33]. Several studies identified the role of RANKL in the acceleration of migration and metastasis of cancer cells. The RANKL/RANK axis is a key molecular link between progestins and epithelial carcinogenesis. The inhibition of RANKL may be considered a putative therapeutic approach for breast cancer. These effects seem to be mediated, at least in part, by the IKK-α–NF-κB signaling pathway [29]. In addition, RANKL/RANK appears to induce epithelial to mesenchymal transition (EMT), migration and invasion through the activation of NF-κB and the upregulation of Snail and Twist in breast cancer cells [31].

The essential role of the RANKL/RANK signaling pathway in the early stages of tumorigenesis was shown in a recent study describing the response of a RANK-positive luminal progenitor (LP) subset (present in BRCA1 mutation carriers) to progesterone-induced RANKL as responsible for the transformation of LPs into basal-like breast tumor cells. The same study suggests denosumab as an effective treatment in the prevention of breast tumorigenesis [34].

Furthermore, the same signaling pathway has been identified as a regulator of the development of resistance to targeted therapies and the proliferation and expansion of cancer stem cell populations [35,36], especially in ERB2+ tumors [9].

## 4. Role of RANKL in the Survival, Resistance Development and Metastatic Capacity of Breast Cancer Cells

Many studies have revealed the key role of the RANKL/RANK axis in chemotherapy resistance for BC patients. The authors of a recent study identified that a secreted form of RANKL was not present in cells exposed to RANKL. It has been shown that the soluble form of RANKL promotes the metastatic capacity of tumor cells in bones. In addition, the authors suggest that a negative feedback loop leads to continuous RANK activation in ER+HER2- BC cells. However, this also leads to ER and PR downregulation, in addition to increased resistance to chemotherapy, potentially providing a second driver of cellular quiescence to anticancer therapy. Therefore, a RANKL-rich background in postmenopausal women may contribute to ER loss and resistance to chemotherapy in ER+ tumors, through continuous RANK activation [35]. The inhibition of RANK signaling has been shown to act as a tumor cell differentiation therapy in BC, depleting the cancer stem cells’ population and reducing recurrence and metastasis [36].

Finally, stem cell properties are directly connected to the process of the EMT phenotype promoting their metastatic capacity. The characteristics of such a mesenchymal transition have been demonstrated in cancer cells that are transfected with RANK. The overexpression of this protein leads to the elevated expression of different transcription factors (SNAIL, TWIST, ZEB1 AND ZEB2) and mesenchymal markers such as vimentin, fibronectin and/or N-cadherin [37,38].

By recognizing the implication of RANKL in the survival, resistance development, EMT phenotype and metastatic capacity of breast cancer cells, the question arising is in which cell types RANKL protein can be expressed. The main supply of production in mice was found to be the tumor-infiltrating CD4+ FoxP3+ regulatory lymphocytes [32], which increased in ERBB2-positive tumors [39].

Breast cancer cells can also produce RANKL and normal cells in metastatic sites, such as osteoblasts in bone [30,40,41,42,43]. RANKL has also been reported to trigger the migration of human epithelial cancer and melanoma cells expressing RANK in a concentration-dependent manner, thus implying a chemotactic mechanism of action in bones [30]. The same molecular mechanism seems to interpret the enhanced osteolysis observed in the context of HIV-1 infection where the HIV-1-infected human macrophages promote the migration of osteoclast precursors via the secretion of RANKL [44]. Moreover, the impaired regulation of RANK/RANKL/OPG systems and excessive osteoclast-mediated osteolysis have been reported to play an important role in promoting bone metastasis [45]. Breast tumor cells can secrete many different cytokines or factors (e.g., IL-1β, IL-6, IL-8, IL-11, IL-17, M-CSF, TNFα, parathyroid hormone-releasing protein (PTHrP), prostaglandin E_2_ (PGE_2_)) that cause increased RANKL production by stromal cells in the bone microenvironment and osteoblasts [46,47]. OPG, the normal decoy receptor for RANKL, is also downregulated via PTHrP, further upregulating the RANKL-to-OPG ratio in skeletal metastases. Osteoclastic resorption results in an increased release of transforming growth factor (TGF-β) and other factors of the bone matrix, promoting the increased proliferation of cancer cells. A vicious cycle is thus established promoting bone destruction and metastasis [42,45,48].

## 5. RANKL/RANK Pathway and Members of the ERBB Family in Breast Carcinogenesis

The association between the RANK signaling pathway and different members of the ERBB family has also been highlighted in different studies [28,32,49]. The ERBB family includes four distinct types of transmembrane tyrosine kinase receptors (EGFR, HER-2, HER-3 and HER-4), which are responsible for the establishment of several intracellular signaling pathways associated with normal cellular function and cancer cell development [50,51]. The RANK pathway is involved in ERBB2-positive carcinogenesis [50]. ERB2 (HER-2) appears to be overexpressed in 20% of BC tumors and is associated with poor prognosis for patients. The well-described downstream mediator of RANK, NF-kB and has also been correlated with cell proliferation, survival, invasion and resistance to agents against ERBB2 in ERBB2-positive BC [52,53,54,55].

Once NF-kB is activated, it triggers the nuclear translocation of IKK in ERBB2-positive cancer cells. This results in increased cell proliferation and the stimulation of cyclin D1 mRNA (CCND1) and protein expression levels, in addition to the self-renewal of tumor-initiating cells (TICs), thus acting on secondary tumor generation. Interestingly, anti-ERBB2 agents may induce the activation of NF-kB, enhancing the oncogene addiction of ERBB2-positive cells to NF-kB, implying that NF-kB initiation could represent a putative resistance mechanism to anti-ERBB2 agents in ERBB2-positive BC patients [56]. Thus, the combination of anti-ERBB2 agents either with inhibitors against proteasome or NF-kB decreases NF-kB activation through the inhibition of IkB degradation and may represent a novel therapeutic approach to treat RANK-expressing ERBB2-positive BC [57].

Although the importance of NF-kB activation in ERBB2-positive BC is highlighted, the exact mechanism of activation has not been entirely clarified. Despite the fact that PI3K/Akt signaling is a primary pathway for tumor development, in ER-negative/ERBB2-positive BC cells, ERBB2 has been shown to stimulate NF-kB through the canonical pathway independently of the PI3K axis. Moreover, IKKa, which is essential in the noncanonical pathway, was identified to have a key role in NF-kB activation and not IKKb. IKKb is crucial in the canonical signaling pathway [49].

A novel molecular pathway for BC tumorigenesis in ER-negative/ERBB2-positive and RANK-expressing BC tumor cells has also been proposed [57]. The activation of ERBB2 dimers leads to an interaction with GAB2 and promotes the formation of a multiprotein complex, including the RANK receptor. The members of the ERBB family interact through GAB2 with the RANK receptor and modulate the NF-kb signaling response, the main signaling pathway downstream of RANK. NF-kB dimer activation is mediated by the canonical pathway [58].

The primary role of IKKa is the phosphorylation of IkB, thus triggering its proteasomal degradation. The activated form of NF-kB translocates to the nucleus, which operates as a transcription factor and induces target gene expression, enhancing ERBB2 expression and generating an NF-kB/ERBB2 feedback loop. This may represent a resistance mechanism against anti-ERBB2 agents. Additionally, nuclear-translocated members of ERBB may act as coactivators of NF-kB, thus enhancing the proliferation and survival of cancer cells by triggering cyclin D1 expression [55,59,60,61,62]. Although this model can partly explain the resistance development to the ERBB2 treatment of ERBB2-positive patients, further studies are needed to elucidate the underlying mechanism.

Recently, it was reported that the RANKL/RANK pathway may also be implicated in the prognosis of ER+ERB2- breast cancer. It is known that ER+ERB2- RANK-overexpressing BC cells have a staminal and mesenchymal phenotype, with a decreased proliferation rate and decreased susceptibility to chemotherapy and hormone therapy (HT). It has been shown that continuous RANK pathway activation in ER+ERB2- cells induces a negative feedback mechanism, inducing ER and PR downregulation and increased resistance to HT [35]. Consequently, the effectiveness of RANKL inhibition in improving ER+ERB2- breast cancer prognosis has to be evaluated in future studies.

## 6. Clinical Trials with Denosumab

Clinical studies have tried to evaluate the beneficial effect of denosumab, alone or in combination, on patients with early-stage or metastatic breast cancer in order to identify the role of denosumab alone (ClinicalTrials.gov Identifiers: NCT03324932, NCT03691311, NCT02613416, NCT01419717). Moreover, the main outcome measures for other studies is the combinatorial effect of denosumab with other interventions such as surgery (NCT02900469), radiotherapy of hormone therapy (NCT02366130) and/or denosumab plus calcium and vitamin D (NCT03629717). Although the majority of studies in Table 1 are in a recruiting status, several studies have already produced results. In the study with ClinicalTrials.gov Identifier NCT01077154, denosumab did not improve disease-related outcomes for women with high-risk early breast cancer [63]. The findings of a multivariate analysis with ClinicalTrials.gov NCT01545648 revealed bone marrow (BM) and disseminated tumor cells’ (DTCs) positivity as an independent risk factor for disease-free survival (DFS), particularly in luminal A/B BC patients. This might be a novel criterion for the identification of patients most likely to benefit from additional adjuvant therapy, possibly including bisphosphonates (denosumab) [64]. Another study (NCT00089661) on patients with nonmetastatic breast cancer and low bone mass receiving therapy with an adjuvant aromatase inhibitor (twice-yearly administration of denosumab) led to significant increases in BMD over 24 months at trabecular and cortical bone [65]. The study with ClinicalTrials.gov Identifier NCT00104650 revealed that among patients with elevated uNTx despite ongoing IV BP therapy, denosumab normalized urinary N-telopeptide (uNTx) levels more frequently than the continuation of intravenous (IV) bisphosphonate (BP) therapy [13]. In the same study on patients with prostate-cancer-related bone metastases and increased urine N-telopeptide, despite IV BP treatment, denosumab normalized uNTx levels more frequently than ongoing IV BP [66]. Table 1 represents clinical studies with denosumab for BC patients with early or metastatic disease.

## 7. RANKL/RANK Signaling Pathway and Tumor Immunomodulation

The essential role of RANKL/RANK signaling in both the stimulation of the immune system (lymph-node and T- and B-cell development) and inhibition of the immune system (generation of regulatory T-cells and stimulation of T-cell tolerance) is well known [67,68]. However, the role of RANK–RANKL signaling in tumor immunology is not yet completely clear. Data from preclinical studies indicate that RANKL inhibition can selectively block medullary thymic epithelial cells (mTECs) and suppress central tolerance while maintaining T-cell generation, potentially providing a therapeutic benefit centrally promoting antitumor immunity. Khan et al. identified that RANKL inhibition led to a transient blockade of central T-cell tolerance, producing enhancement of antitumor immune response in melanoma [69].

Moreover, apart from establishing central tolerance to tumors, RANKL/RANK axis plays a crucial role within the tumors by possibly contributing to the development of a tolerogenic immune microenvironment. As it has already been reported, RANKL has been expressed in tumor-associated macrophages (TAMs) and tumor-infiltrating lymphocytes (TILs) [28,69]. RANKL acts as a chemoattraction factor for TAMs recruited in the tumor microenvironment (TME) and promotes tumor growth, angiogenesis and metastasis [70]. More specifically, the RANKL/RANK axis in M2 type of macrophages regulates chemokines’ secretion, which enhances T regulatory cells (Treg) proliferation [71].

Conversely, Treg lymphocytes produce RAKNL. A vicious cycle is generated during which an increase of Treg cells is induced by M2 TAMS residing in the tumor microenvironment, while at the same time, RANKL, which is produced by Tregs, recruits more M2. The immunosuppressive microenvironment fosters tumor expansion and the development of metastases. In an animal model, tumor-infiltrating Tregs have been identified to initiate metastases via the RANKL/RANK signaling pathway [31]. Consequently, it can be speculated that RANKL/RANK inhibition may impede this process. Anti-RANKL agents increase DC survival and activate the secretion of several cytokines such as IL-1, IL-6, IL-12. Therefore, cytokines appear to trigger the differentiation of CD4+T-cells into TH1 cells, which may increase CD4+T-cell responses. RANK is also expressed on natural killer (NK) cells, thus contributing to immunosurveillance. To this end, inhibition of the RANKL/RANK axis led to a decrease of Tregs in a mouse model with type 1 diabetes [72]. Contrariwise RANKL inhibition in mouse models of lung adenocarcinoma that responded to anti-RANKL agents identified only modest changes in the tumor microenvironment (TME), including the nonspecific depletion of tumor T-cells [73]. RANK is expressed on NK cells and has been shown to inhibit their antitumor reactivity in patients with hematologic malignancies [74,75]. In a recent study application of anti-RANKL has been shown to partially restore NK cell antitumor reactivity [76].

Two recent clinical studies trying to identify the effect of denosumab in lymphocyte cell lines reported contradictory results. In the former study [77] no change in T and B lymphocyte subpopulations has been observed, whereas in the latter stimulation of both T- and B-cells was demonstrated following denosumab administration [78]. Two current studies are in progress to identify the effect of denosumab on the systemic immunity and local immunologic TME. The PERIDENO study (NCT03532087) [79], a prospective randomized study, investigates the efficacy of neoadjuvant denosumab in postmenopausal women with early BC HER2-negative in combination with AC-T chemotherapy. In the second study, ISS 20,177,041 evaluates the use of denosumab as monotherapy in cervical cancer [80].

## 8. Coinhibition of RANKL and Immune Checkpoint Inhibitors (ICIs) as a Putative Therapeutic Strategy

Recognizing that the RANKL/RANK pathway has a multifactorial effect in the immune system and might be modulated in several ways, the antitumor immunity led to evaluating the potential role of RANKL inhibitors in improving the efficacy of ICIs in the treatment of cancer [81,82].

Inhibitors of immune checkpoint proteins such as cytotoxic T-lymphocyte-associated protein-4 (CTLA-4) and antiprogrammed death-1 (PD-1) or its ligand (PDL-1) were evaluated in several tumor types such as melanoma, head and neck squamous carcinoma, NSCLC, renal cell and urothelial carcinoma, gastric cancer and Hodgkin’s lymphoma with satisfactory results [83]. However, there remains a proportion of patients who do not attain clinical benefit from anti-PD1 or anti-CTLA-4 monotherapy or the combination of these two ICIs, due to primary resistance [84].

In 2016, a case of a patient with metastatic melanoma who received denosumab for bone metastases while he was concomitantly treated with the anti-CTLA-4 antibody ipilimumab demonstrated a dramatic improvement with the combination of these two drugs. Intriguingly, the authors pointed out that anti-CTLA-4 and anti-RANKL monoclonal antibodies, used as monotherapy, had poor antimetastatic results. It was postulated that the successful synergistic effect of anti-CTLA-4 and anti-RANKL must have been based on the lymphocytes’ action, as treatment was totally ineffective in mice lacking all lymphocytes and in mice specifically depleted of natural killer cells [85].

Moreover, another study identified that the inhibition of RANKL augmented the antimetastatic efficacy of anti-PD-1/PD-L1 monoclonal antibodies (MoABs) and enhanced growth suppression in the prostate, colorectal cancer and melanoma cell lines in mouse models [86]. The surface receptor PD-1 (CD279), first discovered on a murine T-cell hybridoma, is homologous to CD28 and its primary role is associated with immunosuppressive activity. It is expressed by most circulating T-cells upon stimulation through the T-cell receptor (TCR) complex or exposure to cytokines and TGF-β. By interfering with the early TCR/CD28 axis, PD-1 signaling leads to decreased cytokine secretion, transcription factors and cell cycle progression. Moreover, PD-1 activity is only identified during T-cell activation, as its signal can only be transduced during TCR-dependent signaling [87].

Following these reports, Afzal and Shirai retrospectively evaluated the efficacy of combined therapy of the RANKL inhibitor denosumab and ICIs in patients with metastatic melanoma. Although a statistically significant effect of denosumab on overall survival (OS), progression-free survival (PFS) and overall response rate (ORR) was not observed, the patients receiving denosumab ameliorated overall considering their poor prognostic parameters [88].

Furthermore, as checkpoint inhibitors are used to resensitize the immune system against tumor cells, it is speculated that patients may benefit when these therapies are combined with immune stimulatory substances. Three recent trials are ongoing: the trials with ClinicalTrials.gov Identifier NCT03161756 (Evaluation of Denosumab in Combination with Immune Checkpoint Inhibitors in Patients with Unresectable or Metastatic Melanoma (CHARLI)) [89], NCT03280667 (Denosumab and Pembrolizumab in Clear Cell Renal Carcinoma (KEYPAD)) [90] and NCT03620019 (Denosumab + PD-1 in Subjects with Stage III/IV Melanoma). These studies aim to evaluate the combination of denosumab with ICIs in patients with unresectable or metastatic melanoma and renal cell carcinoma, respectively. In another impending research, the POPCORN study, the combination of preoperatively administered nivolumab (a PD-1-blocking antibody) with denosumab will be studied in resectable NSCLC [90] (Table 2).

## 9. Mechanisms Involved in the Treatment with RANKL and ICIs

Today, the underlying mechanisms of this synergistic action of RANKL and ICIs remain unclear. A recent study highlights the efficacy of combinatorial treatment with agents against RANKL and anti-CTLA-4 antibodies through the analysis of TILs, metastatic potential of tumor cells and tumor growth in different models using a plethora of neutralizing antibodies and gene-targeted mice [91].

Furthermore, in solid tumors, RANKL inhibitors appear to improve the efficacy of anti-CTLA-4 moAbs. The efficacy of optimal combinatorial treatment of anti-CTL4 and anti-RANKL agents was identified through the mouse IgG2a isotype, which has been previously highlighted to selectively deplete intratumoral Tregs. Despite this, this combination depended on the presence of activating Fc receptors and lymphocytes (particularly NK and CD8+ T-cells), whereas anti-RANKL agents alone did not require Fc receptors. The combination therapy resulted in increased T-cell infiltration into solid tumors with the subsequent increase of their effector function [86]. Possible explanations of the synergistic action of anti-RANKL and anti-CTLA4 and/or anti-PD1 therapies is that anti-RANKL antibodies suppress PDL1 expression in the tumor microenvironment, which is considered as a possible resistance mechanism to anti-PD1 treatments. Myeloid cells derived from multiple myeloma peripheral blood, when cultured with RANKL and M-CSF, obtained an immunosuppressive phenotype, suppressing cytotoxic T-cell proliferation and activity. To this effect, IDO and IL-10 seemed to play a significant role. This effect was reversed by the addition of anti-PDL1 [92]. In mouse tumor models, RANKL expression was identified on a subset of cytotoxic CD8 T-cells expressing high levels of PD1. The PD1 blockade increased RANKL expression on tumor-infiltrating T-cells. Concomitant anti-RANKL and the anti-PD1 blockade were effective only in the presence of NK cells and IFNγ, while the antitumor activity of the triple blockade of RANKL, PD1 and CTLA4 required the presence of T-cells and IFNγ [93] (Table 2). The role of T-cells and cytokines is also confirmed by studies showing that anti-RANKL agents added to the immune checkpoint inhibitor compared with an immune checkpoint inhibitor alone resulted in increased CD8+ T-cell infiltrates and the increased production of IFNγ and IL-2by Th1-T-cells in tumors. Additionally, RANK expression has been observed in M2 immunosuppressive tumor-associated macrophages, suggesting its potential immunosuppressive role [94]. Of note is that almost all CD8+RANKL+ T-cell tumor-infiltrating lymphocytes coexpressed PD-1 in mouse tumors [93] (Figure 1).

Consistent with these findings, the dual targeting of PD-1 and RANKL with a bispecific antibody enhanced antitumor activity compared to combined anti-RANKL and anti-PD-1 treatment in mice models [95]. Nevertheless, further studies are needed to explain the effects of RANKL/RANK inhibition on the tumor microenvironment. Therefore, this putative strategy may be beneficial and may enhance the effect of immunotherapy against cancer.

## 10. Conclusions

The RANKL/RANK axis is closely associated with tumorigenesis initiation and progression in breast cancer. RANKL/RANK inhibition, besides its role in preventing and managing SREs, seems to constitute a novel and promising therapeutic approach either as a prevention strategy or as adjuvant therapy in RANK-positive BC. Interestingly, the impaired regulation of the RANKL/RANK axis seems to possess a critical role in promoting bone metastases through the secretion of cytokines by BCs, which cause increased RANKL production by stromal cells in the bone microenvironment. Furthermore, RANKL/RANK signaling pathway inhibition has been identified to modulate the immune environment and indicated to improve the efficacy of anti-CTLA-4 and anti-PD1 monoclonal antibodies against solid tumors. In more detail, the RANKL/RANK axis has been identified to induce primary resistance to treatment with immune checkpoint inhibitors through different mechanisms. Administration of the anti-PD1 blockade leads to the increased expression of RANKL on tumor-infiltrating T-cells and reduced cytokine production by Th1 cells. RANKL/RANK signaling induces the release of immunomodulatory factors that cause reduced NK cell antitumor reactivity and immunosuppression. RANKL inhibition with denosumab as an immunomodulatory modality enhancing the immune response and rendering resistant tumors responsive to immune therapies appears as an emerging therapeutic strategy currently being evaluated in clinical and experimental trials.

Since a functional and physical association between ERBB2 family members and the RANK receptor has been shown, clinical trials of combinatorial administration of immune checkpoint inhibitors and anti-RANKL agents in breast cancer, especially in RANK-expressing ERBB2-positive patients, are needed. In order to clarify the mechanism of the advantages of this double-pathway inhibition strategy in cancer treatment, interesting issues to be addressed could include:(1)The evaluation of change in chemokine production (TNF-alpha, interleukins, IFN gamma) both in serum and in the TME under the conditions of RANKL/RANK inhibition.(2)A change in intratumoral T-cell (CD4, CD8 and Treg) numbers and function to determine the intratumor ratio of effectors to regulators.(3)A change in myeloid cell (M1/M2 macrophage, myeloid-derived suppressor cell (MDSC), DC) numbers and function.(4)Functional markers of myeloid cells expressing RANK. The identification of the main subpopulations of myeloid-derived suppressor cells (MDSCs) expressing RANK may serve as biomarkers for identifying patients who would benefit more from this therapeutic approach.

## Figures and Tables

**Figure 1 ijms-21-07570-f001:**
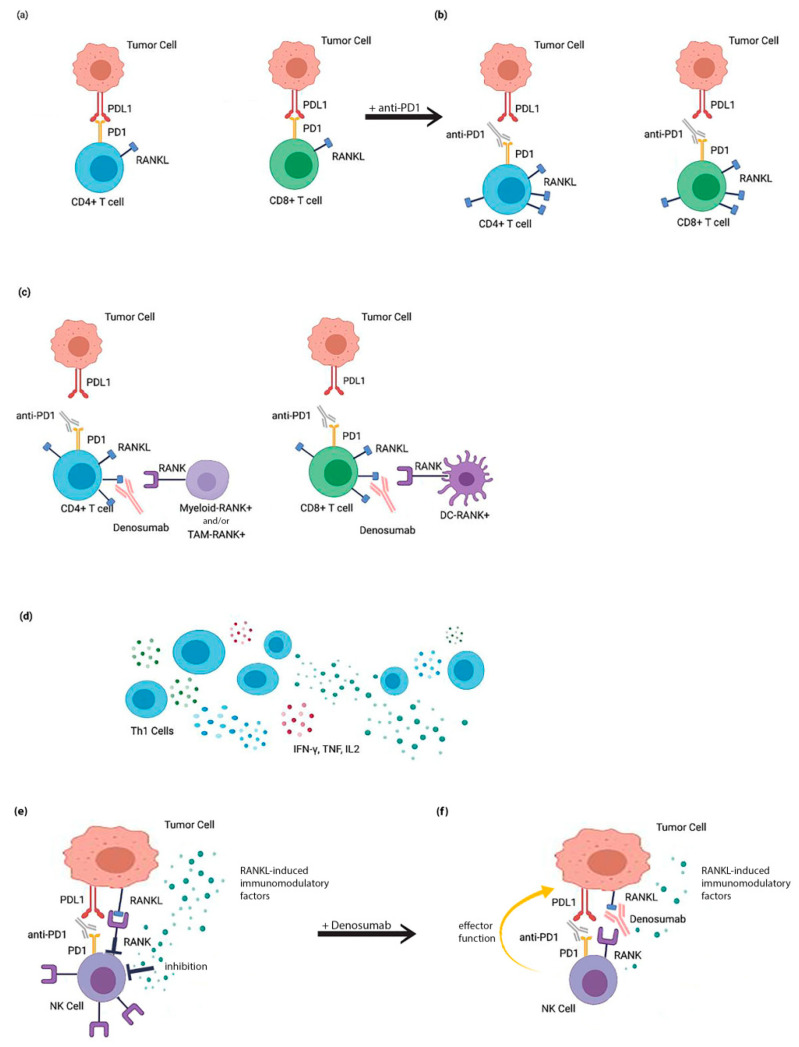
Coinhibition of receptor activator of nuclear factor-κB ligand (RANK) and immune checkpoint inhibitors (ICIs) in order to overcome primary resistance in RANK-positive breast cancer (BC) cells. (**a**) PD1–PDL1 interaction between RANK (+) BC cells and RANKL(+) CD4+ and CD8+ T-cells. (**b**) Administration of the anti-PD1 blockade leads to the increased expression of RANKL on tumor-infiltrating T-cells. (**c**) The combined administration with anti-PDI plus anti-RANKL leads to a decrease in immunosuppression mediated by RANK+ myeloid/dendritic/tumor-associated macrophage (TAM) cells. (**d**) Combinational therapy leads to an increase of T-cell infiltration and cytokine production (IFN-γ, TNF, IL2) by Th1 cells. (**e**) RANK–RANKL signaling induces the release of immunomodulatory factors by RANKL(+) BC cells that directly inhibit natural killer (NK) cell reactivity and induce additional RANK expression on their surface. RANK induces inhibitory signals to NK cells upon interaction with RANKL expressed by BCs. (**f**) The use of denosumab blocks the RANK–RANKL interaction. As a result, there is a reduced release of RANKL-induced immunomodulatory factors. Additionally, denosumab prevents inhibitory RANK signaling into NK cells and provokes enhanced NK cell antitumor reactivity, which acts synergistically with anti-PD1 immunomodulatory action.

**Table 1 ijms-21-07570-t001:** Clinical studies with denosumab for breast cancer patients.

Study Number	Intervention	Phase	Cancer Type	Status
NCT03324932	Denosumab	III	Breast cancer	Recruiting
NCT02900469	Denosumab and surgery	I	Breast cancer	Recruiting
NCT01077154	Denosumab, placebo	III	Breast cancer	Terminated, has results [63]
NCT02366130	Ra-223 dichloride, denosumab, hormone therapy	II	Breast cancer	Active, not recruiting
NCT01545648	Denosumab	II	Breast cancer	Terminated, has results [64]
NCT00556374	Placebo, denosumab, nonsteroidal aromatase inhibitor therapy, zoledronic Acid	III	Breast cancer	Active, not recruiting, has results
NCT00091832	Denosumab, IV bisphosphonates	II	Breast cancer	Completed
NCT03691311	Denosumab	I	Breast cancer	Recruiting
NCT00089661	Placebo, AMG 162/denosumab	III	Breast cancer	Completed, has Results [65]
NCT03629717	Denosumab, calcium, vitamin D	I	Breast cancer prevention	Completed
NCT02613416	Denosumab	II	Breast cancer	Recruiting
NCT01419717	Denosumab	III	Bone metastasis	Completed
NCT02051218	Denosumab (reduced/standard dose)	III	Metastatic disease	Recruiting
NCT02721433	Pamidronate, denosumab, zoledronic acid	IV	Metastatic disease	Active, not recruiting
NCT00950911	AMG 162 (denosumab)	III	Bone metastasis	Completed
NCT00104650	AMG 162	II	Bone metastasis	Completed, has results [13,66]

NCT: national clinical trial; Ra-223: radium 223; AMG 162: denosumab.

**Table 2 ijms-21-07570-t002:** Clinical studies with denosumab and checkpoint inhibitors in solid tumors.

Study Number	Intervention	Phase	Cancer Type
NCT03161756	Denosumab, nivolumab, ipilimumab	I/II	Melanoma
NCT03280667	Pembrolizumab plus denosumab	II	Renal cell carcinoma, clear cell metastatic kidney cancer
NCT03620019	Denosumab, pembrolizumab, nivolumab	II	Melanoma

NCT: national clinical trial.

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
