# Peer review of "The Role of the RANKL/RANK Axis in the Prevention and Treatment of Breast Cancer with Immune Checkpoint Inhibitors and Anti-RANKL"

_ijms, 2020, doi:10.3390/ijms21207570_

Round 1

Reviewer 1 Report

The link between RANK / RANKL signaling and the tumor immune environment is interesting and also this is an interesting review in the context of today's progress in treatment with immune checkpoint inhibitors.
There is only one minor suggestion. If combination of RANK / RANKL targeted therapies and immune checkpoint inhibitors are potentially useful in RANKL + HER2 + breast cancer, why are clinical trials preceded for other cancers (melanoma and renal cancer)? If there is a rationale that performed in those carcinomas, it is considered informative to add it.

Author Response

There is only one minor suggestion. If combination of RANK / RANKL targeted therapies and immune checkpoint inhibitors are potentially useful in RANKL + HER2 + breast cancer, why are clinical trials preceded for other cancers (melanoma and renal cancer)? If there is a rationale that performed in those carcinomas, it is considered informative to add it.

AUTHOR RESPONSE: Initially, we would like to thank the reviewer for his kind comments. About the minor suggestion, to our knowledge, there is not any particular rationale that performs in those carcinomas. Preclinical experiments shows that the combination of anti-RANKL and anti-CTLA4 mAbs significantly reduces experimental mouse melanoma and lung metastases (References 86,87). Since the RANK / RANKL / OPG signaling pathway is also present in the breast and immunotherapy is already used in breast cancer, we recommend in the current review the co-administration of ICI with denosumab for the treatment of breast cancer.

Reviewer 2 Report

In this review, the Authors recapitulated the role of the receptor activator of nuclear factor-κB (RANK) and RANK ligand (RANKL) in breast cancer development and treatment with both immune checkpoint inhibitors and anti-RANKL therapeutics.

The issue is interesting, however it is difficult to understand the meaning of many sentences throughout the manuscript, diverse typos are also present (for instance, see: “Therefore, in a RANKL rich background, like in post-menopausal women may contribute for ER loss and resistance to chemotherapy in ER+ tumors, independently of high RANK expression [32]. Furthermore, inhibition of RANK axis acts as a differentiation therapy in BC, depleting the cancer stems cells population and reducing recurrence and metastasis [33]. These actions have been inhibited by denosumab, confirming the role of RANKL in the process of stem cells’ renewal in ERB2-positive cells [7]”. Moreover, the section should be written in a logic and clear manner to appreciate the sense of the consecutive paragraphs (for instance, why the authors put the following sentences in the “RANKL/RANK signaling pathway”: “In 2009, a human monoclonal antibody against RANKL, denosumab, was developed to inhibit the interaction between RANK and its ligand RANKL [7,8]. In 2011 the drug was approved for osteoporosis treatment and bone metastases in the carcinomas such as breast and prostate [9]. Since then, denosumab was widely used in breast cancer (BC) patients with metastatic disease and was shown to be equal or superior than zoledronic acid in holding or preventing skeletal related events (SREs) [10][11][12][13]”. Moreover, the Authors wrote at the end of the Conclusion section: “we propose trials”. The Authors should comment the previous and the current trials addressing further directions, however it is unusual to conclude the manuscript with a list of proposed trials. Who should perform these trials? The Authors? Others that have stated a similar point of view? This remains a puzzling question.

Next, the discussion would benefit of the following recent studies:

https://doi.org/10.3390/ijms21186723 

https://doi.org/10.3390/ijms21093154

 https://doi.org/10.3390/ijms20112795

 https://doi.org/10.3390/ijms150917130 

Author Response

Initially, we would like to thank the reviewer for his detailed remarks and fruitful comments that will be beneficial for the current manuscript. Below we provide our detailed answers to each comment one-by-one to ensure the clarity of our statements

The issue is interesting, however it is difficult to understand the meaning of many sentences throughout the manuscript, diverse typos are also present (for instance, see: “Therefore, in a RANKL rich background, like in post-menopausal women may contribute for ER loss and resistance to chemotherapy in ER+ tumors, independently of high RANK expression [32]. Furthermore, inhibition of RANK axis acts as a differentiation therapy in BC, depleting the cancer stems cells population and reducing recurrence and metastasis [33]. These actions have been inhibited by denosumab, confirming the role of RANKL in the process of stem cells’ renewal in ERB2-positive cells [7]”.

AUTHOR RESPONSE: The meaning of sentences has been clarified and typos have been corrected.  

 Moreover, the section should be written in a logic and clear manner to appreciate the sense of the consecutive paragraphs (for instance, why the authors put the following sentences in the “RANKL/RANK signaling pathway”: “In 2009, a human monoclonal antibody against RANKL, denosumab, was developed to inhibit the interaction between RANK and its ligand RANKL [7,8]. In 2011 the drug was approved for osteoporosis treatment and bone metastases in the carcinomas such as breast and prostate [9]. Since then, denosumab was widely used in breast cancer (BC) patients with metastatic disease and was shown to be equal or superior than zoledronic acid in holding or preventing skeletal related events (SREs) [10][11][12][13]”.

AUTHOR RESPONSE: The above paragraph was put at the end of the first entity referring to the general description of the RANK/RANKL signaling pathway to introduce the use of denosumab (RANKL inhibitor) in osteoporosis and bone metastases.

In the following entities, after analyzing the role of RANK/RANKL in the normal mammary gland, the role of anti-RANKL treatment as a new approach in preventing and treating breast cancer and metastases, alone or in combination with other treatments and its mode of action is evaluated.

Moreover, the Authors wrote at the end of the Conclusion section: “we propose trials”. The Authors should comment the previous and the current trials addressing further directions, however it is unusual to conclude the manuscript with a list of proposed trials. Who should perform these trials? The Authors? Others that have stated a similar point of view? This remains a puzzling question.

AUTHOR RESPONSE: The above sentence has been rephrased. In their opinion, the authors meant that such studies (for example, performed by pharmaceutical companies) would be with clinical interest and hopeful outcomes for the patients.

Next, the discussion would benefit of the following recent studies:

https://doi.org/10.3390/ijms21186723 

https://doi.org/10.3390/ijms21093154

 https://doi.org/10.3390/ijms20112795

 https://doi.org/10.3390/ijms150917130 

AUTHOR RESPONSE: We thank the reviewer for the interesting papers. The suggested recent studies have been included in the text, as we considered them useful for the improvement of our manuscript. We also added the relevant references (6, 7, 16, 44).

Round 2

Reviewer 2 Report

The Authors attempted to address the comments of the Reviewer.